# Basidiomycetous Yeast, *Glaciozyma antarctica,* Forming Frost-Columnar Colonies on Frozen Medium

**DOI:** 10.3390/microorganisms9081679

**Published:** 2021-08-07

**Authors:** Seiichi Fujiu, Masanobu Ito, Eriko Kobayashi, Yuichi Hanada, Midori Yoshida, Sakae Kudoh, Tamotsu Hoshino

**Affiliations:** 1Bioproduction Research Institute, National Institute of Advanced Industrial Science and Technology (AIST), 2-17-2-1, Tsukisamu-higashi, Toyohira-ku, Sapporo 062-8517, Hokkaido, Japan; nora.yeast@gmail.com (S.F.); zero-ethanal.bk@docomo.ne.jp (M.I.); 8acs1135@mail.tokai-u.jp (E.K.); hanada.yuichi@gmail.com (Y.H.); 2Graduate School of Science, Hokkaido University, N10 W8, Kita-ku, Sapporo 060-0810, Hokkaido, Japan; 3School of Biological Sciences, Tokai University, 1-1-1, Minaminosawa 5, Minami-ku, Sapporo 005-0825, Hokkaido, Japan; 4Hokkaido Agricultural Research Center, NARO, Hitsujigaoka 1, Toyohira-ku, Sapporo 062-8555, Hokkaido, Japan; midori@affrc.go.jp; 5Biology Group, National Institute of Polar Research, 10-3, Midori-cho, Tachikawa, Tokyo 190-8518, Japan; 6Department of Life and Environmental Science, Faculty of Engineering, Hachinohe Institute of Technology, Obiraki 88-1, Myo, Hachinohe 031-8501, Aomori, Japan

**Keywords:** cold adaptation, ecological strategy, frost resistance, ice-binding proteins (IBPs), polysaccharide, unfrozen water

## Abstract

The basidiomycetous yeast, *Glaciozyma antarctica*, was isolated from various terrestrial materials collected from the Sôya coast, East Antarctica, and formed frost-columnar colonies on agar plates frozen at −1 °C. Thawed colonies were highly viscous, indicating that the yeast produced a large number of extracellular polysaccharides (EPS). *G. antarctica* was then cultured on frozen media containing red food coloring to observe the dynamics of solutes in unfrozen water; pigments accumulated in frozen yeast colonies, indicating that solutes were concentrated in unfrozen water of yeast colonies. Moreover, the yeast produced a small quantity of ice-binding proteins (IBPs) which inhibited ice crystal growth. Solutes in unfrozen water were considered to accumulate in the pore of frozen colonies. The extracellular IBPs may have held an unfrozen state of medium water after accumulation in the frost-columnar colony.

## 1. Introduction

Even in the continental Antarctic region, which represents one of the coldest places in the world, various eukaryotic microorganisms are known to exist. Isolated basidiomycetous fungi in this region occurred in yeast-form except for alien species [1], and basidiomycetous yeasts were dominant among culturable fungal strains in the McMurdo Dry Valley [2] and the ice-free area of the Lützow-Holm Bay [3,4,5,6]. Basidiomycetous yeasts have belonged in *Pucciniomycotina*, *Ustilaginomycotina*, and Tremellomycetes in *Agricomycotina*. Tremellomycetes and Agaricomycetes (known species were filamentous form) were detected in soils near glacial lakes in the McMurdo Dry Valley [7] and Victoria Land [8] by DNA metabarcording, and Tremellomycetes were the dominant taxa in these areas. DNA metabarcording suggested that *Coprinellus* and *Pleurotus* (both in Agaricomycetes) were included in dominant fungal taxa on rocks in the polar desert of the Ellsworth Mountains [9].

Previously, we described that ecophysiological strategies for freezing tolerance of filamentous fungi differed among taxa in in the Northern Hemisphere [10]. Filamentous basidiomycetes such as *Typhula ishikariensis* secreted abundant ice-binding proteins (IBPs) (>95% of total extracellular proteins) [11] to keep the extracellular environment unfrozen. This fungus also produced extracellular polysaccharides (EPS) which covered the hyphae to inhibit the diffusion of IBPs so that thermal hysteresis activity may be maintained [10]. However, in basidiomycetous yeasts, there are few reports on frost resistance.

Several cold-adapted yeasts, including basidiomycetes, could grow under freezing conditions [12,13] because they were exceptionally well adapted to cold environments where they thrived in combinations of several stressful conditions [8,14,15]. Among basidiomycetous yeasts, trehalose and glycerol are well-known frost protectants in yeasts, and trehalose accumulated in the cytoplasm in *Mrakia* spp. and *Leucosporidium* spp. to reduce the freezing point of the cytoplasm [16,17]. Biosynthesis pathways of glycerol, trehalose, polyamines, and aromatic amino acids in *M. blollopis* were accelerated under freezing temperatures [18]. Polyamines and aromatic acids are recognized to function to support cell growth under low temperatures. However, low temperatures caused glycerol accumulation instead of trehalose in *M. psychrophila* [19].

IBPs or their activities were recorded in basidiomycetous yeasts in Pucciniomycotina, such as *Glaciozyma antarctica* (syn. *Leucosporidium antarcticum* [20,21,22], *Leucosporidium* sp. [23], *L. creatinivorum* [24], *Psychromyces glacialis* (‘*Rhodotorula svalbardensis*’) [25], *Rhodotorula glacialis* [20]; Tremellomycetes such as *Goffeauzyma gastrica* (syn. *Cryptococcus gastricus*) [24]. However, *Mrakia* spp. in Tremellomycetes could not produce IBPs [20], and draft genome analysis of *M. blollopis* suggested that strain SK-4 did not have IBP genes [26]. Therefore, *Mrakia* spp. were less frost tolerant to freeze-thaw cycles among basidiomycetous yeasts isolated from Antarctica [24].

In this article, we isolated the basidiomycetous yeast, *G. antarctica*, which formed irregular frozen colonies like frost columns at subzero temperatures and elucidated their growth mechanism on frozen media. Our results indicated that *G. antarctica* had unique physiologic characteristics against freezing stress to adapt to continental Antarctica.

## 2. Materials and Methods

### 2.1. Sample Collection and Fungal Isolation on Frozen Media

Various terrestrial samples such as soil, mosses, and algal mats were collected from eight ice-free areas in the Coasts of Sôya and Prince Harald, and Lützow-Holm Bay, East Antarctica from December 2006 to February 2007 (Figure 1). Samples were collected using disposable γ-ray sterilized spatulas or autoclaved tweezers and were kept in γ-ray sterilized plastic bags. Sample types are shown in Appendix A, and collected samples were stored at −20 °C before isolating fungi that grew on frozen potato dextrose agar (PDA, Becton, Dickinson and Company, Sparks, MD, USA) plates cultured according to Hoshino et al. [27]. Samples, each weighing 100 mg, were placed on PDA plates and frozen at −80 °C for 24 h. Frozen plates were transferred to −1 °C for six months. Fungi that grew on frozen PDA were transferred and maintained on PDA slants at 4 °C.

### 2.2. Phylogenic Analysis

DNA was extracted from 2–5 fungal isolates, each with similar morphological characteristics and grown on frozen PDA, according to the protocol of Isoplants II Kit (Wako Pure Chemical Industries, Osaka, Japan). The internal transcribed spacer (ITS) region of genomic rDNA was amplified with the primer pairs ITS1F [28] and ITS4 [29]. LSU rDNA D1/D2 domain of 5 yeast isolates that formed frost-columnar colonies on PDA were also amplified with the primers ITS1F and NL4 [30]. PCR products were purified using QIAquick PCR Purifications Kit (Qiagen, Hilden, Germany) and sequenced on an ABI PRISM 3100 Genetic Analyzer (Applied Biosystems, Foster City, CA, USA) using primers ITS1F (ITS region of tested isolates) and ITS1F, ITS4, NL1 and NL4 (ITS region and D1/D2 domains of *G. antarctica* formed frost-columnar colonies).

Multiple alignments of ITS region and D1/D2 domain sequences were performed using CLUSTAL W (http://clustalw.ddbj.nig.ac.jp/) (access on 26 July 2021) and manually adjusted. A phylogenic tree was constructed by the Maximum parsimony method and using the program MEGA X [31] with bootstrap values based on 1000 replications.

### 2.3. Morphological Analyses on Frost-Columnar Colonies

One loopful of cells of *G. antarctica* S4B was inoculated on a frozen PDA plate including 0.05% (*w*/*v*) red food color (Kyoritsu Foods Co., Tokyo, Japan) and was incubated at −1 °C for two months. Three plates were used for this experiment in triplicate. Surface features of frost-columnar colonies were observed by scanning electron microscopy (SEM). Frost-columnar colonies were lyophilized and coated with platinum-palladium using JFC-1100 Ion Sputter (JEOL, Tokyo, Japan). They were examined using JSM-T330A SEM (JEOL, Tokyo, Japan) operating at 15 kV.

### 2.4. Chemical Analyses of EPS

Strain S4B was inoculated on frozen or unfrozen PDA plates (regular, 1/2, 1/5, and 1/10 nutrient conditions). Three plates were used for each experimental condition in triplicate. Frost-columnar colonies on PDA were thawed and suspended in water. Yeast cells were removed by centrifugation (15,000× *g*, 20 min at 2 °C). HPLC determining the molecular mass of polysaccharides in the supernatant was performed according to Yoshida et al. [32], and sugar composition was analyzed following Matsuyama et al. [33].

### 2.5. IBP Activity in Frost-Columnar Colonies and Culture Broth

The frozen colony of *G. antarctica* S4B on PDA was thawed at 4 °C and suspended in 10 mL of 0.1 M ammonium bicarbonate (pH 7.9). Yeast cells and EPS were removed by centrifugation (15,000× *g*, 20 min at 2 °C). The supernatant was washed three times by 0.1 M ammonium bicarbonate (pH 7.9) and concentrated by the ultrafiltration to SDS-PAGE and immunoblotting by anti-*Typhula ishikariensis* IBPs antibody [34]. SDS-PAGE followed the methods of Laemmli [35]. Immunoblotting analysis was made using the ECL Western Blotting Detection System (Cytiva, Tokyo, Japan).

Fifty milliliters of each of *G. antarctica* S4B, *Coprinus psychromorbidus* [36], and *T. ishikariensis* BRB cultures were prepared by inoculating two loopfuls of cells of *G. antarctica* S4B or two mycelial discs (5 mm in diam), cut from the margin of an actively growing colony on a PDA plate, into potato dextrose broth and then incubating for 3 months at −1 °C without shaking. Yeast cells were removed by centrifugation (15,000× *g*, 20 min at 2 °C), and mycelia were also removed by filtration.

The IBP activity of the concentrated supernatant was examined on the development of ice crystals under a Leica DML 100 photomicroscope (Leica Microsystems AG, Wetzlar, Germany) equipped with a Linkam LK600 temperature controller (Linkam, Surrey, UK) [37]. Each experiment was tested in triplicate.

### 2.6. Statistical Analysis

Relationships between frost-columnar colonies and sampling sites or sample types were analyzed using ANOVA and Tukey’s HSD test (significant for *p* < 0.05).

## 3. Results and Discussion

### 3.1. Fungal Growth on Frozen Media

Several filamentous fungi and yeasts grew from terrestrial samples inoculated at −1 °C on frozen PDA plates (Table 1). Of 277 samples examined, 166 showed mycelial growth, and frost-columnar colonies occurred in 37 samples (details of isolates are shown in Appendix A). There was no relationship between sample type and fungal occurrence.

Some colonies on frozen PDA were irregular in shape and resembled frozen “marshmallows” in appearance (Figure 2A). We named these frozen colonies “frost-columnar colonies”, and all these colonies were included in white yeasts. Five isolates of these yeasts among frost-columnar colonies were identified as *G. antarctica* since they were highly homologous to the ITS region and D1/D2 domains of *G. antarctica* CBS 5942^T^ (Appendix A), and the phylogenetic tree was a similar morphology as previously recorded [38]. *G. antarctica* was exclusively isolated from them and formed the same colonies in aseptic cultures of frozen PDA (Figure 2C). We used the S4B strain isolated from soil in East Onglu Is. for later experiments, and we did not observe that other fungal isolates without *G. antarctica* formed frost-columnar colonies in our experimental condition.

Thawed colonies were highly viscous due to a large amount of carbohydrates, which were identified by the phenol sulfuric acid method. HPLC analysis revealed that the carbohydrates were polymerized hexoses (30–100 molecules) composed of ca. 69% D-glucose, 30% D-galactose, and 1% D-mannose.

### 3.2. The Structure of Frozen Colonies

During the growth of the yeast on frozen PDA containing red food coloring, the pigment was concentrated in the frozen colony (Figure 2C). This result suggested that frozen colonies accumulated unfrozen water containing red food coloring and other solutes in PDA. The surface of the colony was frosted and pale pink, but inside the colony, pigmentation became intense to the interface with PDA (Figure 3A). The colony was lyophilized to investigate its structure by SEM. The surface had honeycomb-like structure (Figure 3B), and the center was porous like a sponge (Figure 3C). There were large holes in the bottom (Figure 3D,E). Yeast cells were suspended in the hole by EPS or attached to the wall of the hole (Figure 3F).

### 3.3. Cell Growth and EPS Production on Frozen Media

The colonies of *G.*
*antarctica* on frozen PDA looked like frost columns, while on unfrozen PDA, the colonies were typical of yeast colonies (Figure 4B). While cell growth on frozen PDA culminated in 10^8^ cells/plate after 17 weeks, the maximal cell growth was attained after 5 weeks on unfrozen PDA (Figure 4A). EPS content in the frost-columnar colonies decreased to ca. 1/5 when colonies were kept at −1 °C for 1 year (material not intended for publication: Seiichi Fujiu, Graduate School of Science, Hokkaido University, Sapporo, Japan, Master thesis entitled “Studies on environmental adaptation of Antarctic basidiomycetous yeast formed frost-columnar colonies under frozen conditions” in Japanese, 2010). Therefore, EPS produced by the yeast also acted as a storage material for their growth.

*G**laciozym antarctica* is known to produce EPS abundantly [39]. Cultural conditions had a significant influence on the EPS productivity of *G. antarctica*. EPS productivity on frozen PDA was 50 to 100 times higher than that on unfrozen PDA, and EPS productivity was highest on 1/2 frozen PDA (Figure 5). Nutrients in the culture decreased by half on 1/10 frozen PDA after 20 weeks.

Basidiomycetous yeast *Mrakia blollopis* changed their morphology under different nutrient conditions [40]; *M. blollopis* showed yeast cells on eutrophic media and hyphal cells on oligotrophic media. Filamentous growth allows fungi to explore a more excellent surface search for water and nutrients and makes it possible to form aggregates to retain water and nutrients in unconsolidated and sandy soils [8]. On the other hand, *G. antarctica* kept the yeast-form under the conditions tested. On frozen PDA, the cell growth of *G. antarctica* was delayed, but maximal cell numbers attained on frozen PDA were the same as those on unfrozen PDA. These results suggested that *G. antarctica* adapted well to the frozen environment but that their physiological characteristics were not osmophilic or xerophilic.

### 3.4. IBP Activity in Frozen Media

Extracellular IBP activity of *G. antarctica* was found from culture broth [20,21]. Expressions of IBP genes in *G. antarctica* PI12 were induced by subzero temperatures under an unfrozen condition [21,41]. Arctic diatoms in sea ice avoid freezing through thermal hysteresis by IBP and EPS [42]. We also observed a distorted hexagonal bipyramidal shape of ice crystals based on IBP activity from extracellular fractions of frost-columnar colonies (Figure 6A). The molecular mass of major proteins in extracellular fractions was >30 kDa (Figure 6B), similar to the molecular mass of Afp4 (one of the isoforms of IBPs) in *G. antarctica* PI12 [22]. This major protein was immunoreacted with the anti-*Typhula ishikarienis* antibody (Figure 6C). Extracellular IBP productivity of *G. antarctica* S4B was less than those of basidiomycetous snow molds, *T. ishikariensis* and *C. psychromorbidus* (Figure 6D).

Both filamentous basidiomycetes produced extracellular IBPs abundantly (>90–95% of total extracellular proteins) to adapt to freezing environments; however, *T. ishikariensis* may inhibit mycelial growth at subzero temperatures when IBPs freely diffuse into the extracellular environment [10,27]. Possibly, IBP molecules are bound with EPS, covering the mycelia of *T. ishikariensis* to protect from freezing damage [10]. Both basidiomycetes, *G. antarctica* and *T. ishikariensis*, produce both EPS and IBPs for frost resistance, but the ratios of EPS to IBPs were contrasting; *G. antarctica* produced a large amount of EPS and a small amount of IBPs to form frozen colonies. While *T. ishikariensis* always develops mycelia, *G. antarctica* exists in the form of yeast. Such a difference in the life form of both fungi implies that they adopted different strategies to adapt to frozen environments. Further investigations on other fungi are necessary to generalize the hypothesis.

## 4. Conclusions

We found that the basidiomycetous yeast, *G. antarctica,* produced a large amount of EPS and developed frost-columnar colonies on frozen PDA. These frozen colonies were porous, and yeast cells were found aligned along the walls of the pores. We considered that unfrozen water containing nutrients was accumulated within the pores of frozen colonies through the capillary phenomenon and the Marangoni effect by EPS, and that yeast cells absorbed nutrients in liquid water for growth. IBP was detected in frozen colonies. IBPs in frozen colonies of *G. antarctica* may keep accumulated water in the liquid state.

*Glaciozyma antarctica* in continental Antarctica and *T. ishikariensis* in the Boreal zone were well adapted to subzero temperatures to utilize EPS and IBPs. However, while *T. ishikariensis* applied these substances to control the micro-environment around their mycelia under supercooling conditions, *G. antarctica* applied these substances to absorb nutrients in liquid water under freezing conditions, indicating that both fungi used the same substances for different purposes.

## Figures and Tables

**Figure 1 microorganisms-09-01679-f001:**
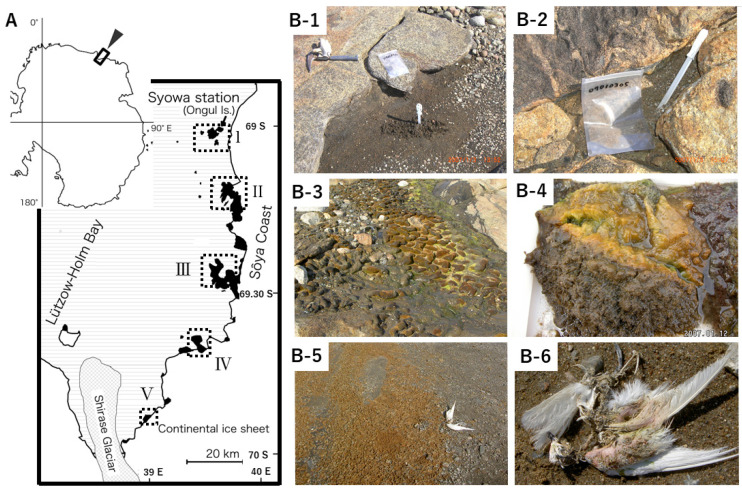
Locations of sampling sites and various terrestrial samples in this study. (**A**) Sampling sites: I: Ongul Is., II: Langhovde, III: Skarvsnes, IV: Skallen, V: Rundvågshetta. (**B**) Example of sampling sites and collected samples: soils in East Ongul Is. (**B**-**1**,**2**), moss colonies in Skarvsnes (**B**-**3**), algal mats from the bottom of freshwater lakes in Skarvsnes (**B**-**4**), dried algal mats (**B-5**), and dead birds near freshwater lakesides in Skarvsnes (**B-6**).

**Figure 2 microorganisms-09-01679-f002:**
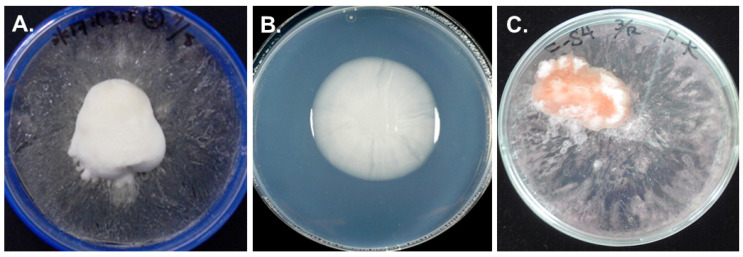
Colonies of *G. antarctica* at subzero temperature. Soil near Hyoga-Ike (glacier pond) in Landvode was inoculated on frozen PDA at −1 °C for 3 months (**A**). S4B strain was isolated from soil in East Onglu Is. Their growth on unfrozen PDA at −1 °C for 1 month (**B**) and frozen PDA, including red food coloring for 2 months (**C**).

**Figure 3 microorganisms-09-01679-f003:**
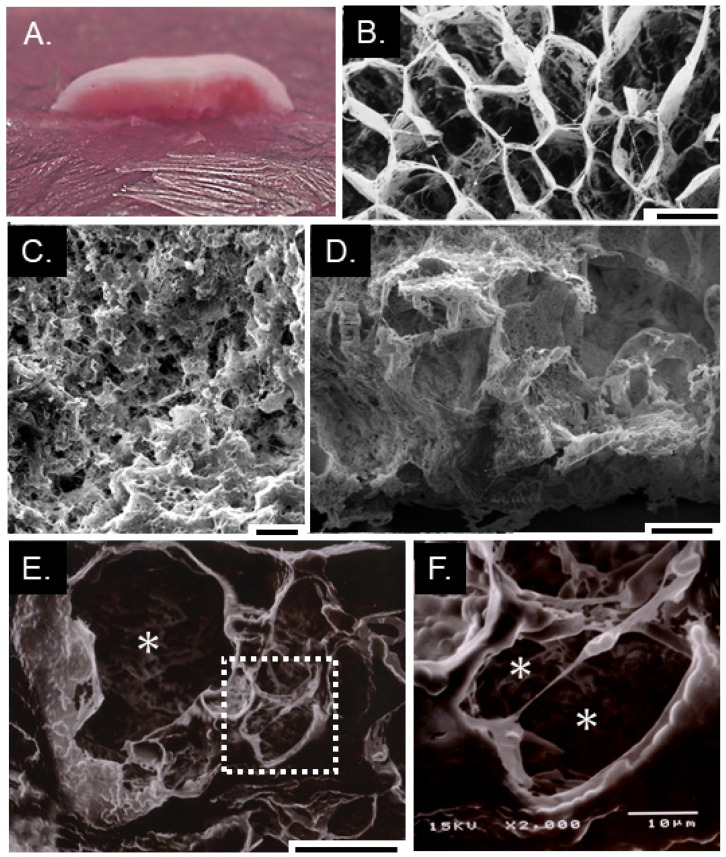
The structure of *G. antarctica* colony on frozen PDA. Cut surface of the colony on frozen PDA containing red food coloring (**A**). The honeycomb structure of the colony surface (**B**). Sponge-like structure of the center (**C**). Longitudinal section of the bottom (**D**,**E**). Close-up view of the rectangular in E. * Holes. Bars 100 μm (**B**–**D**), 50 μm (**E**), and 10 μm (**F**).

**Figure 4 microorganisms-09-01679-f004:**
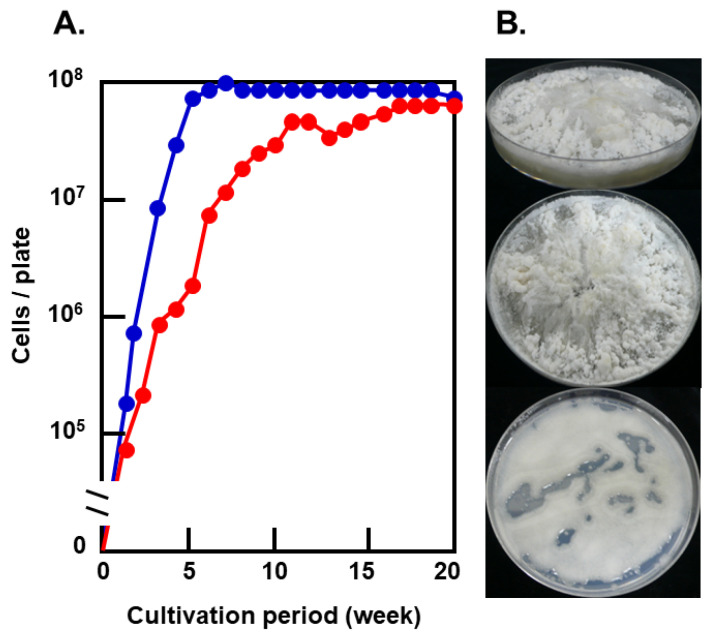
Growth of *G. antarctica* at −1 °C on PDA. (**A**). Growth of *G. antarctica* on unfrozen PDA (blue) and frozen PDA (red). (**B**). Frozen PDA cultures at −1 °C for 20 weeks (top and middle) and unfrozen PDA cultures at −1 °C for 5 weeks (bottom).

**Figure 5 microorganisms-09-01679-f005:**
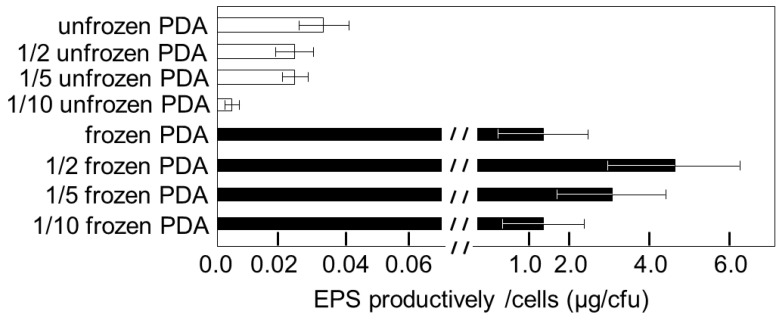
Effect of cultural conditions on extracellular polysaccharide productivity of *G. antarctica* S4B. Cultures were incubated at −1 °C for 20 weeks on frozen PDA or 5 weeks on unfrozen PDA. Error bars indicated SD (*n* = 3).

**Figure 6 microorganisms-09-01679-f006:**
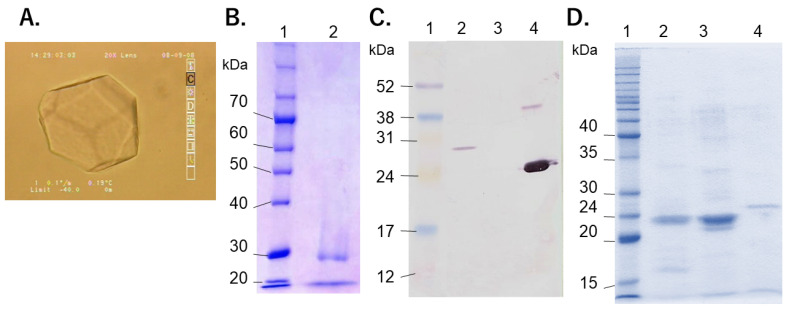
IBPs in both the frozen colony and culture of *G. antarctica* strain S4B. Development of ice crystal in the extracellular fraction from frost-columnar colony (**A**). SDS-PAGE of the extracellular fractions from frost-columnar colony with Coomassie Brilliant Blue staining (**B**) Immunoblotting of the extracellular fractions from frost-columnar colony and SDS-PAGE of culture broths of *G. antarctica* strain S4B and other basidiomycetes (**C**). Molecular marker (lane 1), the extracellular fractions from frost-columnar colonies (lane **B2**, **C2** and **D4**), culture broth of *Pichia pastoris* without *Tis*AFP gene (negative control; lane **B3**), culture broth of *P. pastoris* transformed *Tis*AFP6 gene [43] (positive control; lane **B4**), culture broth of *Coprinus psychromorbidus* (lane **D2**), and culture broth of *T. ishikariensis* BRB (lane **D3**). Forty μL of the extracellular fractions and culture broths were loaded on lanes 2–4, respectively.

**Table 1 microorganisms-09-01679-t001:** The number of fungi that grew on frozen potato dextrose agar, including those that formed frost-columnar colonies.

Locality	No. Samples Examined	No. Samples with Fungal Growth	Samples Producing Frost-Columnar Colonies
Sôya Coast Ongule Islands			
East Ongul Is.	54	36	8
West Ongul Is.	13	9	0
Ongul Kalven Is.	7	0	0
Langhovde	40	26	6
Skarvsnes	125	71	15
Skallen	11	7	1
Rundvågshetta	14	12	6
Prince Harald Coast			
Riiser-Larsen Peninsula	13	5	1
Total	277	166	37

## Data Availability

Master thesis of S.F. were kept in the library, Faculty of Science/Graduate School of Science/School of Science, Hokkaido University, Sapporo, Japan.

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
