# Peer review of "Basidiomycetous Yeast, Glaciozyma antarctica, Forming Frost-Columnar Colonies on Frozen Medium"

_microorganisms, 2021, doi:10.3390/microorganisms9081679_

Round 1

Reviewer 1 Report

Summary:

During this study, the presence and the production of EPS by Glaciozyma antarctica were investigated. In particular, the Authors studied the difference of forming frost column-like colony on frozen medium comparing the unfrozen water. They also highlighted different production of EPS and IBPs between Glaciozyma antarctica and Typhula ishikariensis useful for the frost resistance

Overall assessment:

I think that the idea to assess the presence of Glaciozyma antarctica and state the EPS production in different Antarctic samples is innovative and could be relevant for the overall scientific community.

However, I have some major and minor concerns regarding the manuscript presented by the Authors.

I think that the Authors should greatly revise the abstract omitting results that are not shown in the present work, reshaping the discussion by implanting physiological, ecological and biotechnological applications of this work. Moreover, clear and impressive conclusions should be included because seem very weak or they are not completely clear to me.

I think that the introduction should be revised because does not provide detailed state of the art of the literature available.

Lines 49 The main goals of this study are not totally clear to me. Please clearly resentence explain your aim.

One of the major concerns regards the sampling strategy. Why did the authors sampled and isolated many different strains and analyzed only G. antarctica among 37 frost column-like colonies?

It is not described how they ensure sterile conditions during sampling.

Line 57. Could you draw a map of sampling? could you provide a detailed table with the isolates and sources? How did u choose the source? please indicate

Line 71. Why did you not consider the reverse primers?

Line 85. Did you statistically test that no relationship occurred?

Line 86 Phylogenetic trees are missing and are fundamental for giving more relevance to the work. You should test more than one molecular marker (ITS).

Line 99 Could you provide results of comparisons of EPS production between the strains isolated?

Line 141. Authors should do statistical tests for assessing differences between samples. How many replicates did you perform? please indicate and show standard deviations on the histograms

Line 151 Why did u use anti-Typhula ishikariensis antibodies? Also, the molecular weight of the protein should be 22-23 kda instead of 30 how do you explain this? Please, explain or cite other work that did this procedure.

Line 167 ref 20 is wrong.

The figure S1 is completely misleading. It is not clear to me the aim to this experiment, which should be better explained. First of all, it is impossible to compare line 3 with the others because comes from another experiment. I think you should provide a detailed quantification with some programs coupled with statistical tests. In the same experiment, you must include a positive control (the Typhula ishikariensis extracts) and negative control for assessing the differences.

The data of Typhula ishikariensis are completely missing

Replication:

The number of replicates of the analyses is not entirely clear. Authors should indicate replicates analysed.

Author Response

30 July 2021

We are grateful to reviewer 1 for the critical comments and useful suggestions that have helped us to improve our paper, As indicated in the responses that follow, we have taken all these comments and suggestions into account in the revised version of our manuscript.

Comment #1: I think that the Authors should greatly revise the abstract omitting results that are not shown in the present work, reshaping the discussion by implanting physiological, ecological and biotechnological applications of this work. Moreover, clear and impressive conclusions should be included because seem very weak or they are not completely clear to me.

Response: I removed the sentence comparing cell growth of G. antarctica and T. ishikariensis. I focused on explaining the mechanism and function of frost column-like colonies for cell growth under subzero temperatures.

Comment #2: I think that the introduction should be revised because does not provide detailed state of the art of the literature available.

Response: Corrections were made to the introductory paragraphs on DNA metabarcoding in Antarctica, cryoprotectants and subzero productivity of basidiomycetous yeasts, and IBP of basidiomycetous yeasts.

Comment #3: Lines 49 The main goals of this study are not totally clear to me. Please clearly resentence explain your aim.

Response: The purpose of this article is to describe the cell growth of basidiomycetous yeast under subzero temperature. For this reason, the last sentence of this section was modified: “Our results indicated that G. antactica had unique ecophysiological strategies against freezing stress that differed from the filamentous fungus, Typhula ishikariensis.”.

Comment #4: One of the major concerns regards the sampling strategy. Why did the authors sampled and isolated many different strains and analyzed only G. antarctica among 37 frost column-like colonies ?

Response: Thirty-seven frost-colored colonies were obtained, and all of them contained white yeast. Five yeasts were randomly selected from the marbled colonies, and their ITS region sequences showed high homology to G. anatrctica. The above is described in lines 143-145 of the revised manuscript.

Comment #5: It is not described how they ensure sterile conditions during sampling.

Response: Details of the sampling method can be found in lines 71-72.

Comment #6: Line 57. Could you draw a map of sampling? could you provide a detailed table with the isolates and sources? How did u choose the source? please indicate

Response: Sampling map was added to the revised manuscript as new Figure 1. Sample type and isolates were listed in new Table S1 and S2, respectively.

Comment #7: Line 71. Why did you not consider the reverse primers?

Response: Since it was possible to identify the isolates at the genus level only by sequencing from one of the primers, no further experiments were conducted. In response to the suggestion by referee 1 that the study was sufficient, information on isolates without frost-columnar colonies was removed from the text, and the list of isolates is shown in new Table S2.

Comment #8: Line 85. Did you statistically test that no relationship occurred?

Response: Relationships between frost-columnar colonies and sampling sites or sample types were analyzed using ANOVA and Tukey’s HSD test. The above was described in lines 127-129 of the revised manuscript.

Comment #9: Line 86 Phylogenetic trees are missing and are fundamental for giving more relevance to the work. You should test more than one molecular marker (ITS).

Response: Phylogenic tree based combined ITS region and D1/D2 domains of Glaciozyma including in our isolates were shown in new Figure S1.

Comment #10: Line 99 Could you provide results of comparisons of EPS production between the strains isolated?

Response: We did not have EPS productivities of our isolates without G. antarctica.

Comment #11: Line 141. Authors should do statistical tests for assessing differences between samples. How many replicates did you perform? please indicate and show standard deviations on the histograms

Response: Three plates were used for each experimental condition in triplicate. The above was described in line 105 in the revised manuscript. SD was added in new Figure 4.

Comment #12: Line 151 Why did u use anti-Typhula ishikariensis antibodies? Also, the molecular weight of the protein should be 22-23 kda instead of 30 how do you explain this? Please, explain or cite other work that did this procedure.

Response: G. antarctica has isoforms of IBPs with different molecular masses due to glycosylation. Known IBPs of G. antarctica and T. ishikariensis were high homology, and protein in frost-columnar colonies crossed with anti-Typhula ishikariensis AFP. This result suggested that IBP in frost-columnar colonies was known IBP in G. antarctica.

The experimental method and results were shown in lines 116-125 and 205-213 in the revised manuscript, respectively.

Comment #13: Line 167 ref 20 is wrong.

Response: I changed reference to [10] in line 217, the revised manuscript.

Comment #14: The figure S1 is completely misleading. It is not clear to me the aim to this experiment, which should be better explained. First of all, it is impossible to compare line 3 with the others because comes from another experiment. I think you should provide a detailed quantification with some programs coupled with statistical tests. In the same experiment, you must include a positive control (the Typhula ishikariensis extracts) and negative control for assessing the differences.

Response: Our purpose of previous Figure 1 was the presence of IBP in frost-columnar colonies. The experimental method and results were described in the revised manuscript (please check Response #13). Positive and negative controls were added in revised Figure 5.

Comment #15: The data of Typhula ishikariensis are completely missing

Response: The data of T. ishikariensis was already published in the new ref. [10]. Therefore, the data related to T. ishikariensis was used in the discussion of the revised manuscript.

Comment #16: The number of replicates of the analyses is not entirely clear. Authors should indicate replicates analysed.

Response: The number of replicates in each experiment was described in Material and Methods in the revised manuscript.

              I believe the manuscript has been improved satisfactorily and hope it will be accepted for publication in Microorganisms.

              Best regards

                                                                     Sincerely yours,

                                                                      Tamotsu HOSINO, Ph.D.

Reviewer 2 Report

In this paper, the authors described the growth of the psychrophilic yeast strain Glaciozyma antarctica. The overall work is well written but requires several improvements.

Here are my comments:
Introduction:
I recommend improving the introduction by adding more information than references to introduce the topic in more depth.

Materials and Methods:
Provide data on the temperature and pH of the sampling site, and add info regarding the pH of the culture media. In general, I appreciate the use of references instead of tedious experimental details, but in this case, the materials and methods are nothing more than a list of references; I suggest adding some more experimental details.

Results:
88 to 99 would be easier for the reader if this list of the isolate were reproduced as a table

Conclusion:
where are the conclusions?

Author Response

30 July 2021

We are grateful to reviewer 2 for the critical comments and useful suggestions that have helped us to improve our paper, As indicated in the responses that follow, we have taken all these comments and suggestions into account in the revised version of our manuscript.

Comment #1: Introduction:
I recommend improving the introduction by adding more information than references to introduce the topic in more depth.

Response: Corrections were made to the introductory paragraphs on DNA metabarcoding in Antarctica, cryoprotectants and subzero productivity of basidiomycetous yeasts, and IBP of basidiomycetous yeasts.

Comment #2: Materials and Methods:
Provide data on the temperature and pH of the sampling site, and add info regarding the pH of the culture media. In general, I appreciate the use of references instead of tedious experimental details, but in this case, the materials and methods are nothing more than a list of references; I suggest adding some more experimental details.

Response: I described sampling site and sample types in new Figure 1 and Table S1, respectively, However, we did not have the data on the temperature and pH of the sampling site. Material and Methods was also revised for your suggestion.

Comment #3: Results:
88 to 99 would be easier for the reader if this list of the isolate were reproduced as a table

Response: I removed this section according to the suggestion of reviewer 1. The list of fungal isolates growing on frozen PDA was shown in Table S2.

Comment #4: Conclusion:
where are the conclusions?

Response: The section of the conclusions was described in the revised manuscript. “Results” and “Discussion” in the previous manuscript were merged. In previous “Discussion”, the sentence corresponding to the summary of the results was moved to the “Conclusion”.

              I believe the manuscript has been improved satisfactorily and hope it will be accepted for publication in Microorganisms.

              Best regards

                                                                             Sincerely yours,

                                                                     Tamotsu HOSHINO, Ph.D.

Round 2

Reviewer 1 Report

I am grateful to the Authors for addressing my criticisms. However, I have still some concerns about their work that I reported below:

Comment #1: I think that the Authors should greatly revise the abstract omitting results that are not shown in the present work, reshaping the discussion by implanting physiological, ecological and biotechnological applications of this work. Moreover, clear and impressive conclusions should be included because seem very weak or they are not completely clear to me.

Response: I removed the sentence comparing the cell growth of G. Antarctica and T. ishikariensis. I focused on explaining the mechanism and function of frost column-like colonies for cell growth under subzero temperatures.

Authors should improve the scientific relevance of the work and find some potential applications with the concentrations of EPS produced by G. antarctica.

In some parts, the papers must be more detailed in some paragraphs extending some concepts. Moreover, in the conclusion paragraph, they should not show any references but just highlight the take-home message that is still not clear to me.

Comment #2: I think that the introduction should be revised because does not provide detailed state of the art of literature available.

Response: Corrections were made to the introductory paragraphs on DNA metabarcoding in Antarctica, cryoprotectants and subzero productivity of basidiomycetous yeasts, and IBP of basidiomycetous yeasts.

Please highlight these corrections

Comment #3: Lines 49 The main goals of this study are not totally clear to me. Please clearly resentence explain your aim.

Response: The purpose of this article is to describe the cell growth of basidiomycetous yeast under subzero temperature. For this reason, the last sentence of this section was modified: “Our results indicated that G. antactica had unique ecophysiological strategies against freezing stress that differed from the filamentous fungus, Typhula ishikariensis.”

These goals do not clearly explain the aims of the work. The word ecophysiological is completely misleading because in the work you did not cover this aspect.

Comment #4: One of the major concerns regards the sampling strategy. Why did the authors sampled and isolated many different strains and analyzed only G. antarctica among 37 frost column-like colonies ?

Response: Thirty-seven frost-colored colonies were obtained, and all of them contained white yeast. Five yeasts were randomly selected from the marbled colonies, and their ITS region sequences showed high homology to G. Antarctica. The above is described in lines 143-145 of the revised manuscript.

I think to demonstrate G. antarctica is a producer of IBPs authors should compare different strains isolated or other taxa including Typhula ishikariensis  as a positive control in the same experimental conditions. The results and references showed by the authors are misleading and difficult to compare among each other.

This is fundamental for demonstrating what the authors state in the conclusion paragraph.

Comment #10: Line 99 Could you provide results of comparisons of EPS production between the strains isolated?

Response: We did not have EPS productivities of our isolates without G. antarctica.

As I described above. I think that it is a fundamental point to highlight in your work.

Comment #11: Line 141. Authors should do statistical tests for assessing differences between samples. How many replicates did you perform? please indicate and show standard deviations on the histograms

Response: Three plates were used for each experimental condition in triplicate. The above was described in line 105 in the revised manuscript. SD was added in new Figure 4.

Is this SE???? how did you perform statistical analyses with them?

Author Response

2 August 2021

We are grateful to reviewer 1 for the critical comments and valuable suggestions that have helped us to improve our paper. As indicated in the responses that follow, we appreciate the additional comments and apologize for causing you trouble.

Comment #1-1A: Authors should improve the scientific relevance of the work and find some potential applications with the concentrations of EPS produced by G. antarctica.
Response: I agree with referee 1. However, the primary purpose of this paper is to elucidate the role of EPS in the growth of basidiomycetous yeast in a frozen environment. Unfortunately, we do not have any results on industrial applications of EPS from G. antarctica.
There has been massive research and applications over the years on the industrial use of polysaccharides produced by microorganisms. Therefore, it would be premature to describe any industrial application from the fundamental results of our study. This journal is a general topic on microbiology. Therefore, we believe that the publication of fundamental research results is acceptable in "Microorganisms."

Comment #1-1B: In some parts, the papers must be more detailed in some paragraphs extending some concepts. Moreover, in the conclusion paragraph, they should not show any references but just highlight the take-home message that is still not clear to me.
Response: I removed references in "Conclusion" and added the following sentence highlighting the take-home-message in lines 247-251, the newly revised manuscript.
"Glaciozyma antarctica in continental Antarctica and T. ishikariensis in the Boreal zone were well adapted to subzero temperatures to utilize EPS and IBPs. However, while T. ishikariensis applied these substances to control the micro-environment around their mycelia under supercooling conditions, G. antarctica had them absorb nutrients in liquid water under freezing conditions, indicating that both fungi used the same substances for different purposes.".

Comment #2: I think that the introduction should be revised because does not provide detailed state of the art of literature available.
Response: Corrections were made to the introductory paragraphs on DNA metabarcoding in Antarctica, cryoprotectants and subzero productivity of basidiomycetous yeasts, and IBP of basidiomycetous yeasts.
Comment #2-1: Please highlight these corrections
Response: It was cited that references [4-6] of the dominant of culturable fungal species in Lützow-Holm Bay (lines 34). DNA metabarcording research [7-9] in continental Antarctica was cited and described their outline related to basidiomycetous yeasts in lines 34-39. Basidiomycetous yeasts could grow under freezing conditions were described [8, 12-15] (lines 47-49). Trehalose, glycerol, and other substances were known in frost protectants in basidiomycetous yeasts [16,17], and the acceleration of cryoprotectants biosynthesis was described [18, 19] (lines 49-55). Overview of IBPs in basidiomycetous yeasts was described [20-26] (lines 56-62).

Comment #3-1: These goals do not clearly explain the aims of the work. The word ecophysiological is completely misleading because in the work you did not cover this aspect.
Response: It is deplorable that our manuscript was not judged to be an outcome of fungal ecophysiology. In your view, the last sentence of this section was modified: "Our results indicated that G. antactica had unique physiologic characteristics against freezing stress to adapt to the continental Antarctica." in lines 65-66, the newly revised manuscript.

Comment #4-1: I think to demonstrate G. antarctica is a producer of IBPs, authors should compare different strains isolated or other taxa, including Typhula ishikariensis, as a positive control in the same experimental conditions. The results and references showed by the authors are misleading and difficult to compare among each other.
This is fundamental for demonstrating what the authors state in the conclusion paragraph.
Response: It was shown in Figure 6D that the comparison of IBP productivities of G. antarctica S4B, T. ishikariensis BRB, and another IBP producing basidiomycete, Coprinus psychromorbidus were cultured under the same experimental condition. Details of this experimental method and results were described in lines 123-127 and lines 219-221, the newly revised manuscript, respectively. Detail of the sampling method was rewritten in lines 71-72.

Comment #10-1: Line 99 Could you provide results of comparisons of EPS production between the strains isolated?
Response: We did not have EPS productivities of our isolates without G. antarctica.
Comment #10-1: As I described above. I think that it is a fundamental point to highlight in your work.
Response: Unfortunately, in the time frame of this manuscript revision, we could not add new experimental data on EPS productivities of other isolates in frozen media. This study focuses on the formation of frost-columnar colonies of G. antarctica in a freezing environment.

Comment #11-1: Line 57. Is this SE???? how did you perform statistical analyses with them?
Response: I am very sorry. That was a typo in SD. We have corrected the description in the newly revised manuscript.

Best regards

Sincerely yours,

Tamotsu HOSHINO, Ph.D.

Reviewer 2 Report

I find this new version of the manuscript much improved there is only a small problem to be solved:

I am a bit confused by the presence of two totally different 4 figures, please fix the problem and the numbering in the text. 

Author Response

2 August 2021

We are grateful to reviewer 2 for the critical comments and useful suggestions that have helped us to improve our paper. As indicated in the responses that follow, we appreciate the additional comments and are sorry for causing you trouble.

Comment : I find this new version of the manuscript much improved there is only a small problem to be solved:

I am a bit confused by the presence of two totally different 4 figures, please fix the problem and the numbering in the text.

Response: I am very sorry for the duplication of Figure 4. It has been corrected.

              I believe the manuscript has been improved satisfactorily and hope it will be accepted for publication in Microorganisms.

                                                                                   Best regards

                                                                                    Sincerely yours,

                                                                                    Tamotsu HOSHINO, Ph.D.